# Labor unionization and real earnings management: Evidence from labor elections

**Vivek Astvansh** [1,2,3,4]*, **Beibei Wang**[5,6], **Tao Chen** [7], **Jimmy Chengyuan Qu**[7]

**1** Marketing Area, Desautels Faculty of Management, McGill University, Montréal, Québec, Canada, **2** Department of Informatics, Luddy School of Informatics, Computing, and Engineering, Indiana University, Bloomington, Indiana, United States of America, **3** Environmental Resilience Institute, Indiana University, Bloomington, Indiana, United States of America, **4** Dewy Data, Inc., Stockton, California, United States of America, **5** Post-Doctoral Station of Applied Economics, Nanjing University, Nanjing, China, **6** Post-Doctoral Workstation, Bank of Nanjing, Nanjing, China, **7** Divison of Banking & Finance, Nanyang Business School, Nanyang Technological University, Singapore, Singapore

* Vivek.astvansh@mcgill.ca

**Data Availability Statement:** Users can download data from https://www.nlrb.gov/data-on-datagov.

**Funding:** The author(s) received no specific funding for this work.

## Abstract

By exploiting the local randomness in close-call labor elections, the authors find a negative impact of labor unionization at a firm on its real earnings management (REM). The finding suggests a managerial pressure effect of increased labor power. In a local regression discontinuity (RD) analysis, firms that narrowly pass the 50% threshold show a significant decrease in REM, relative to their peers that narrowly fail. This effect is stronger for firms headquartered in right-to-work states and when managers have less pressure to manage earnings. Evidence from a global parametric RD analysis and a multivariate OLS test using industry-level unionization measures confirms the external validity of results in local RD analysis. Overall, the research sheds new light on the economic consequence of labor unionization on employers' accounting decisions.

"If a significant portion of our employees were to become unionized, our labor costs could increase and our business could be negatively affected by other requirements and expectations that could increase our costs, change our employee culture, decrease our flexibility, and disrupt our business"

(Starbucks Corporation in its 2022 annual filing with the U.S. Securities and Exchange Commission, p. 18).

## Introduction

Drives to unionize labor in the United States are on the rise. Since December 2021, high-profile companies—such as Amazon, Apple, Chipotle, Delta Air Lines, Google, Starbucks, and Trader Joe's—and even rail companies and political parties—have been exposed to labor unionization. This trend is unlike any seen in the past. For example, on December 9, 2021, the National Labor Relations Bureau (NLRB) certified the employees at a Starbucks store in

**Competing interests:** The authors have declared that no competing interests exist.

Buffalo, New York, to form a union. Until the end of 2022, the NLRB had certified unions at 133 (that is, 2.05% of) Starbucks Corporation's stores and 3,400 (i.e., 1.4% of) Starbucks employees. These numbers become alarming when one juxtaposes them with a unionization rate of 4.6% among retail workers and 1.2% among food service employees in the United States. The opening quote from Starbucks indicates the threat that a firm foresees because of labor unionization.

In another example, the United Auto Workers' 40-day strike against General Motors (GM) is estimated to have cost the company about $4 billion for the year. This loss made GM managers do a lot of work to do in an attempt to make up losses from the strike. Because the growing power of a firm's labor increases the firm's operating risk ([1, 2]), how to manage the firm's earnings becomes a first-order concern of managers. This concern becomes increasingly important to employers after U.S. President Joe Biden promised to be the most pro-union American president. *Does labor unionization at a firm cause the firm's managers to manipulate earnings*? The current research answers this timely question.

Real earnings management (REM) adjusts earnings toward the target by changing the timing and structure of real activities such as sales, inventory, R&D expenses, advertising, and SG&A (e.g., [3, 4]). Relative to REM, accrual-based management of earnings is limited in scope and more easily detectable by outsiders ([4, 5]). Consequently, REM has become more pervasive for managers especially in the aftermath of the Sarbanes-Oxley Act of 2002 and the Enron scandal in 2001 (e.g., [6–9]). Because a powerful labor union likely places more demands on a firm's management, an increase in union power—on average—threatens the firm's earnings and raises its operational risks (e.g., [1, 2]). Therefore, our main hypothesis is that labor unionization at a firm causes the firm to manage its real earnings *downwards*.

Unlike extant research on labor unionization that uses firm and industry variables to measure labor unionization (e.g., [2, 10]), we sample close-call labor elections recorded in the National Labor Relations Board (NLRB). Empirically, identifying the effect of labor unionization is severely hampered by endogeneity problems. Extant firm and industry measures cannot eliminate biases caused by omitted variables and reverse causality. First, omitted variables, such as performance and investment efficiency, may affect both REM and labor unionization. Second, reverse causality may arise when REM impacts labor unionization. For example, firms' REM may hurt employees' interests and force them to unionize. Therefore, a superior identification strategy is needed to alleviate such problems. Because NLRB labor elections use a passing rule of simple majority where elections with over 50% votes are declared as wins and unions constituted (e.g., [1, 11–13]), we implement a *sharp* (as opposed to fuzzy) regression discontinuity (RD) design to measure the (causal) effect of labor unionization.

Following [14], we measure the level of a firm's REM by the negative sum of abnormal operating cash flow and abnormal discretionary expenses. We also report robustness to two measures that extant REM literature has used ([3, 15]). Using a local RD analysis to exploit locally exogenous variation of labor unionization in close-call labor union elections, we find a significant drop in REM for firms that narrowly passed the 50% voting threshold (vs. firms that narrowly "failed" to pass the threshold). This finding supports our main hypothesis that labor unionization at a firm causes its managers to manipulate real earnings downwards. Such manipulation aims to save operating costs and thus lower the odds that the union would place costly demands on the firm.

We report robustness to different kernels, orders of polynomials, bandwidths, and measures of REM (e.g., [16, 17]). Results of placebo tests support that the baseline result is neither an artifact of data nor occurs by chance. To corroborate our main hypothesis, we conduct a

series of tests, exploiting conditions that suggest heterogeneity in the labor union's and managers' incentives. Specifically, while testing heterogeneity in labor union incentives, we find that the effect exists for firms that were headquartered in states that had adopted the right-to-work (RTW) laws when the election took place. Conversely, we find no significant decrease in REM for narrowly winning (vs. narrowly losing) firms headquartered in states that had not adopted RTW laws. Because RTW-states have weaker workplace protections due to shrinking unions (e.g., [18, 19]), this finding suggests that the effect of winning the labor election comes from increased labor power. For heterogeneity in managerial incentives, we observe that firms decrease REM more when they (1) beat analysts' expectations, (2) have more financial resources, and (3) experience lower corporate risks. These findings suggest the important role of managerial pressure in the effect of labor unionization on REM.

Next, we investigate the channels through which labor unionization impacts REM. We find that each of the changes in (1) abnormal discretionary expenses and (2) abnormal operating cash flow is significantly affected under greater labor unionization. The insight is that labor unionization affects a firm's REM on both dimensions. To provide a comprehensive view of how labor unionization impacts a firm's earnings management decisions, we also examine whether labor unionization affects accrual-based earnings management. We find a negative effect on discretionary accruals, suggesting that managers manipulate accrual-based earnings management downwards under greater labor unionization. Next, we also implement two tests to demonstrate the external validity of the local RD analysis. From global parametric RD analysis, we find that the negative effect of labor unionization on REM remains. Using industry-level proxies of labor unionization (e.g., [2, 10]), we implement a multivariate OLS analysis and find a negative association between labor unionization and REM in a more comprehensive sample of U.S. listed firms. These findings support the external validity of results in the local RD analysis.

Our primary finding—that labor unionization causes managers to manage earnings downwards—contributes to the literature in two ways. First, it adds to the literature on the economic consequences of labor unionization. This literature has documented the impact of labor unionization on a firm's financing, investment, payout policy, and operations (e.g., [12, 20–23]). Using evidence from labor negotiation and labor policy changes, academics have documented how employees affect earnings management (e.g., [24–26]). In an RD design based on the 50% cutoff, prior studies identify the effect of labor unionization on innovation ([11]), asymmetric cost behavior ([12]), bankruptcy risk ([1]), and long-run value of shareholders ([13]). Using the same RD framework, this paper shows a significant decrease in a firm's REM under labor unionization and sheds light on the economic implications of labor unions on corporate earnings management.

Second, the finding contributes to the literature on earnings management by documenting labor unionization as a determinant of REM. According to prior research, a firm's REM can be affected by factors such as financial analysts ([16, 17]), creditor monitoring ([4]), litigation risk ([27]), SEO valuation ([14, 28]), and internal governance ([29, 30]). We add to this strand of literature by documenting labor unionization as a determinant of REM. Because we observe a negative impact of labor unionization on firms' accrual-based earnings management and suggest the simultaneous adjustments of accrual-based and REM under labor unionization, we provide a comprehensive view of how labor unionization affects a firm's earnings management decisions.

The remainder of the paper proceeds as follows. The next section reviews related literature and develops our hypotheses. Subsequently, we describe the sample and methodology, present our main results and the results of heterogeneity tests and supplementary analyses, and conclude the manuscript.

## Related literature and hypotheses

A firm is essentially a nexus of contractual relations among stakeholders (e.g., [31, 32]). As one of the most important stakeholders, employees—especially unionized employees—should impact a firm's decision-making processes. Extant literature has documented that labor unionization impacts a firm's capital structure, investment, bankruptcy risk, cost of capital, production, CEO compensation, dividend payout, and asymmetric cost behavior, thus affecting the firm's value to shareholders and debtholders (e.g., [2, 8, 12, 13, 20–23, 27, 33–36]).

Managers may face greater potential loss in earnings and higher operational risk under labor unionization (e.g., [1, 2]). Extant literature implies that managers may *inflate* earnings to avoid potential negative market reactions related to union pressure because hitting earnings benchmark is important for the firm and its managers (e.g., [7, 8, 37, 38]).

However, recent research has emphasized the direct impacts of union power. Specifically, a powerful union can request higher wages, shorter hours, and more extensive fringe benefits. Further, because of its legal right to strike and enter into collective-bargaining agreements, the union can also increase costs for employers, impeding corporate operations, and discouraging discretionary investments ([11, 39]). In line with this negative view of labor-union power, we presume that managers will deflate earnings through earnings management and thus attempt to lower the operating costs resulting from labor unionization (e.g., [8, 40–42]).

**H₁**. *All else being equal, labor unionization at a firm decreases its real earnings management.*

Next, we examine how the employees' and managers' incentives moderate the impact of labor unionization on REM. Although labor unionization increases labor power, the magnitude of increased power is determined by local regulations. Because right-to-work (RTW) laws are designed to allow non-union members (who do not pay membership fees) to have the same benefits as union members, more employees choose not to join labor unions. As such, shrinking labor unions in states that have adopted these laws (hereafter, RTW-states) may have fewer resources, limited power, and insufficient incentives to bargain for employees (e.g., [1, 11]]). That is, RTW-states have weaker workplace protections. For example, RTW-states are more likely to have lower minimum wages and weaker laws protecting workers from discrimination and harassment, which arise largely from the weakening of union power ([18, 19]). Taken together, labor unionization will have a stronger effect in RTW-states by providing stronger workplace protections. If unionized labor impacts REM, firms headquartered in states that have adopted RTW laws—where workplace protections are constrained—are expected to have stronger needs to deflate earnings through REM. Taking advantage of the RTW adoption, we propose our second hypothesis as follows.

**H₂**. *All else being equal, the effect of labor unionization at a firm on its real earnings management is stronger for firms headquartered in states with more unionization incentives.*

To trade-off between the potential costs and investors' demands for the risk compensation under greater labor unionization (e.g., [1, 2]), managers are more likely to deflate earnings when they face lower pressure on earnings, especially when they beat analysts' expectations, have adequate financial resources, and face lower corporate risks (e.g., [4, 43, 44]). Because a healthy profit profile provides managers with more power to manipulate earnings (e.g., [4, 5]), labor unionization should have a stronger impact on REM for firms that experience lower external pressure. Accordingly, we hypothesize:

**H₃**. *All else being equal, the effect of labor unionization at a firm on its real earnings management is stronger for firms with less external pressure on earnings.*

Because the extent to which labor elections impact REM depends on employees' and managers' incentives, testing $H_2$ and $H_3$ helps corroborate $H_1$.

## Sample selection and methodology

### The sample

We source the labor election data from the NLRB database and the accounting and financial data from Standard & Poor's Compustat–Capital IQ North America Fundamentals Annual (hereafter, Compustat). Our unit of observation is a firm-year-labor election case and our initial sample includes 7,229 such observations. Following extant research that uses NLRB data ([11, 13]), we include in our sample observations that meet the following six criteria. (1) The election must have more than 150 eligible voters. This step decreased our sample from 7,229 to 3,493 observations. (2) We should be able to match the employer's name recorded in the NLRB database to Compustat (sample reduced to 1,494 observations). (3) Real earnings management (REM) measures are available in the year of the election and the year following the election (1,327 observations). (4) The firm must not have experienced another key development (e.g., merger and acquisition, CEO turnover, or shareholder activism activities) in the year of the NLRB election (1,059 observations). (5) The firm must not be in the financial (SIC code 6000–6999) and utility (SIC code 4900–4999) industries (881 observations). (6) Data must be available for all control variables (654 observations).

Thus, our sample contains 654 firm-year observations covering elections from 1989 to 2021. For a firm-year with more than one election, we keep the election with the highest percentage of votes. We winsorize all the continuous variables at the 1st and 99th percentiles. Fig 1 plots the NLRB elections used in the RD analysis. Specifically, the bar chart shows the number of NLRB elections while the line graph depicts the average percentage of votes.

As Fig 1 shows, the number of election cases has dropped gradually since 2010. This trend is consistent with the argument that comprehensive labor protection laws and the adoption of RTW laws have reduced employees' incentives to form labor unions ([11, 13]).

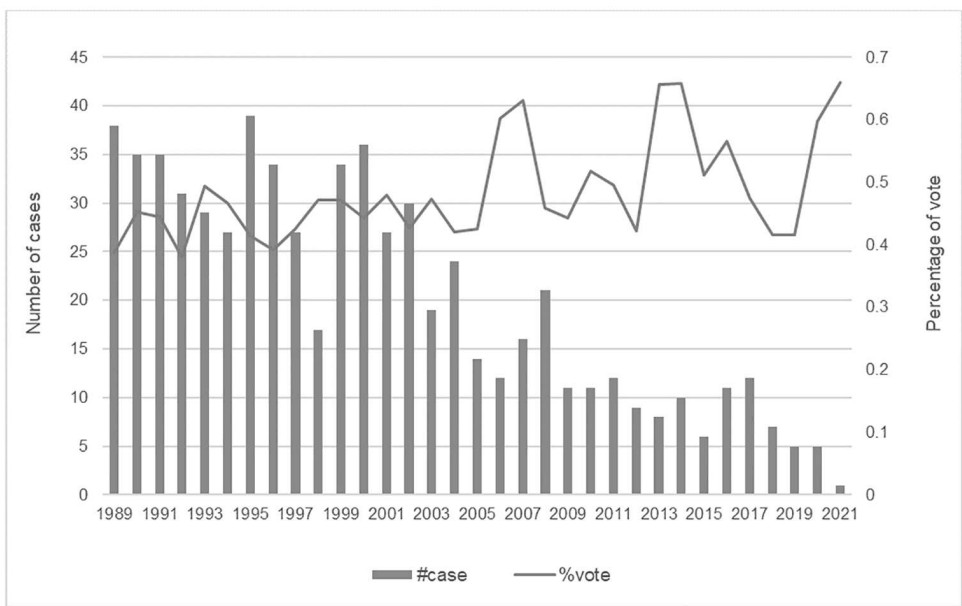

**Fig 1. NLRB cases and percentage of vote in the RD analysis.** *Note*: This figure plots the NLRB cases used in the RD analysis.

## Measures of REM

We first follow [14] and measure REM by the negative sum of abnormal operating cash flow and abnormal discretionary expenses:

$$REM1 = -(REM\_OANCF + REM\_DISX)$$

*REM_OANCF* and *REM_DISX* denote abnormal operating cash flow and abnormal discretionary expenses, respectively. Based on [4], abnormal operating cash flow is the residual of the following equation:

$$\frac{OANCF_{it}}{AT_{i,t-1}} = \alpha_0 + \alpha_1 \frac{1}{AT_{i,t-1}} + \alpha_2 \frac{SALE_{it}}{AT_{i,t-1}} + \alpha_3 \frac{\Delta SALE_{it}}{AT_{i,t-1}} + \varepsilon_{it}$$

*OANCF* denotes the operating cash flow.

Abnormal discretionary expenses (*REM_DISX*) is the residual of the following equation:

$$\frac{DISX_{it}}{AT_{i,t-1}} = \alpha_0 + \alpha_1 \frac{1}{AT_{i,t-1}} + \alpha_2 \frac{SALE_{it}}{AT_{i,t-1}} + \varepsilon_{it}$$

*DISX* is the sum of advertising, R&D, and SG&A expenses.

Considering the role of abnormal production costs in REM ([4, 40]), we follow [9] and construct an alternate measure of REM (*REM2*) by calculating the difference between abnormal production costs and abnormal discretionary expenses (e.g., [16, 17]):

$$REM2 = REM\_PROD - REM\_DISX$$

Abnormal production costs (*REM_PROD*) is the residual of the following equation:

$$\frac{PROD_{it}}{AT_{i,t-1}} = \alpha_0 + \alpha_1 \frac{1}{AT_{i,t-1}} + \alpha_2 \frac{SALE_{it}}{AT_{i,t-1}} + \alpha_3 \frac{\Delta SALE_{it}}{AT_{i,t-1}} + \alpha_4 \frac{\Delta SALE_{i,t-1}}{AT_{i,t-1}} + \varepsilon_{it}$$

*PROD* is the sum of the cost of goods sold and changes in inventory during the year. We do not use the aggregate of *REM_PROD* and *REM_OANCF* because this method can result in double counting ([4, 14, 27]).

Besides, we test robustness using an alternate measure of abnormal operating cash flow (*OANCF_R*) defined by [45]:

$$\frac{OANCF_{it}}{AT_{i,t-1}} = \alpha_0 + \alpha_1 \frac{SALE_{it}}{AT_{i,t-1}} + \alpha_2 \frac{\Delta SALE_{it}}{AT_{i,t-1}} + \alpha_3 \frac{PROD_{it}}{AT_{i,t-1}} + \alpha_4 \frac{DISX_{it}}{AT_{i,t-1}} + \varepsilon_{it}$$

*PROD* is the sum of the cost of goods sold and changes in inventory during the year, and *DISX* is the sum of advertising, R&D, and SG&A expenses. Based on this measure, we construct *REM1_R* by substituting *REM_OANCF* from *REM_OANCFR*.

We estimate each of the above "abnormal" regressions for each industry-year that has more than 15 observations. We identify an industry by two-digit Standard Industrial Classification (SIC) code.

## RD method

The RD analysis uses labor election cases to identify the effect of labor unionization at a firm on its REM. Specifically, we implement a *sharp* RD design by exploiting the NLRB labor elections' passing rule of a simple majority. That is, elections with over 50% votes are declared a win for the labor, and a union is constituted (e.g., [1, 11–13]). NLRB records the results of

labor union elections under the procedures specified by the National Labor Relations Act (NLRA). Firms that narrowly pass or fail the labor elections provide a locally exogenous variation to measure the effect of labor unionization under the RD design (e.g., [1, 11, 13]). Different from firm-level measures from SEC filings or Thomson Reuters Asset4 (e.g., [10, 46]) or industry-level measures from the Unionstats database (e.g., [2, 47]), the RD design provides an ideal identification strategy that rules out the potential endogeneity problem to the most extent. Although [46] deal with the same research question as ours, we use completely different methods and reach different conclusions. Specifically, [46] use firm-level labor union membership data (2002–2016) to measure the union power, while we use the labor election data (1989–2021). Unlike their firm-level measure, our RD design can provide an ideal identification strategy to capture the causal effect of union power. Moreover, contrary to their conclusion that labor unions increase REM, we find a negative impact of labor unions on REM, supporting the managerial pressure effect of increased labor power rather than the managerial incentive effect.

## Main result

### Empirical strategy

To test $H_1$, the following dynamic RD specification estimates the local average treatment effects (LATEs) of labor unionization on REM:

$$\Delta REM_{i,t+1} = \alpha + \beta Unionization_{it} + F_r(X_{it}, \gamma_r) + F_l(X_{it}, \gamma_l) + \varepsilon_{i,t+1} \qquad (1)$$

*REM* denotes the measure of real earnings management (*REM1*) defined by [14] (We use a dynamic form of regression to analyze the difference-in-differences of dependent variables (DVs) in the RD design. Because there is no pre-existing difference in REM measures, using REM measures at $t + 1$ or their changes from $t$ to $t + 1$ as DVs should not change our results. We report robustness to these measures). *Unionization* is an indicator that equals 1 if the labor election received greater than 50% votes (i.e., passed). X denotes the vote margin, which is the difference between the vote rate and the passing threshold (0.5 in this article). $F_r(X_{it}, \gamma_r)$ and $F_l(X_{it}, \gamma_l)$ are polynomials on either side of the zero vote-margin. We select the bandwidth based on [48]'s method of mean squared error. Per our $H_1$, we expect the coefficient of *Unionization* to be negative.

### RD validation tests

The validity of RD relies on two assumptions ([49, 50]). First, the vote share should be smooth around the passing threshold. Such smoothness ensures that the voting is not manipulated. Second, treated firms and their control counterparts around the passing threshold must be similar in firm-specific fundamental characteristics. We test each assumption next.

One can argue that managers of the focal firm may manipulate labor elections. Therefore, we test the continuity of the vote distribution before the analysis. Fig 2 plots the density distribution of union vote shares in labor elections of U.S.-listed firms using the method proposed by [51]. Fig 2 reports no significant discontinuity around the 50% cutoff, meaning that the elections are not systematically manipulated.

To test whether firms that narrowly pass the threshold have similar fundamentals as firms that narrowly fail the threshold, we follow extant RD research ([52]) to select seven firm fundamentals: firm size, leverage, Tobin's Q, return of assets, collateral, sales growth rate, and firm risks. Panel B of Table 1 shows the summary statistics of firm fundamentals. These statistics are similar to those reported in prior research. Panel C of Table 1 reports the results of tests on

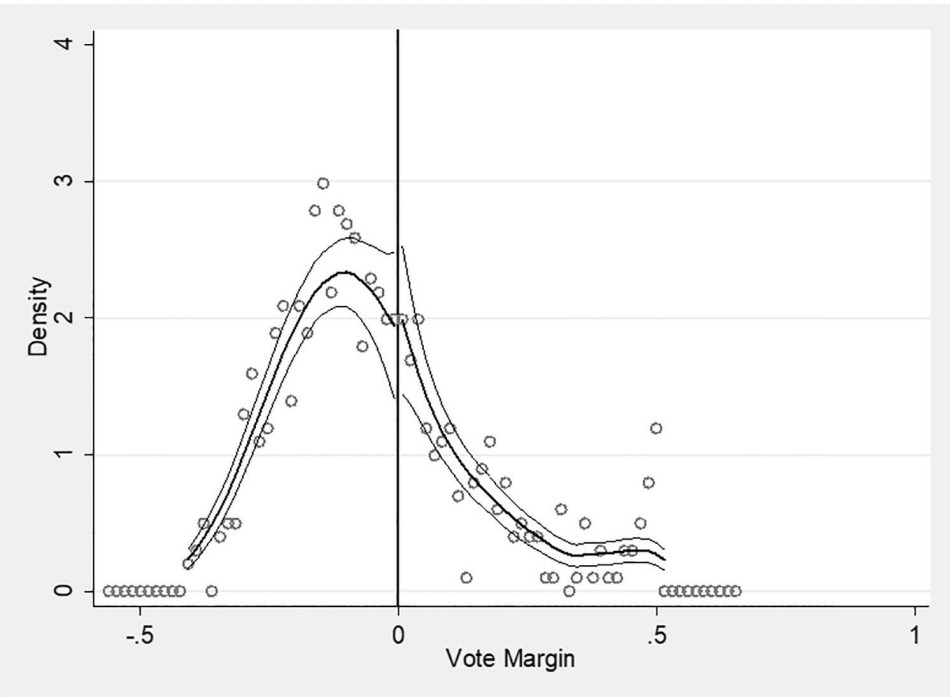

**Fig 2. Density distribution of union vote shares.** *Note*: This figure plots the density distribution of union vote shares in labor elections of U.S. listed firms using the method proposed by [59]. The x-axis shows the vote margin. The y-axis and dots show the density estimates. The solid lines show the fitted density function of the percentage of votes with a 95% confidence interval. The vote margin is the difference between the vote rate and the passing threshold (0.5 in this paper).

firm fundamentals listed in Panel B. We note that values of none of the seven variables change significantly around the 50% cutoff.

Overall, these tests suggest that (1) the votes are not systematically manipulated in the proximity of the 50% threshold; (2) firms on either side of the threshold are not significantly different in fundamentals. These results together confirm that RD is a valid design for our data.

## Graphical analyses

We graph the change in a firm's REM after labor elections. Fig 3 shows regression discontinuity plots using a fitted quadratic polynomial estimate with a 95% confidence interval around the fitted value. It reports a significant drop around the 50% cutoff for the REM measure and thus provides model-free evidence of the *negative* effect of labor unionization at a firm on its REM.

## Main effect using local RD analysis

Next, we conduct a local RD analysis to formally measure the effect of labor unionization at a firm on its REM.

As Table 2 reports, the LATEs of labor unionization are negative and significant at 1% level across different measures of DVs, kernels, and polynomial orders. The interpretation is that relative to a firm that narrowly loses the labor election, a firm that narrowly passes the election decreases its REM. For economic magnitude, firms that narrowly pass the elections, on

**Table 1. Summary statistics.**

**Panel A: REM measures**

|  | N | Mean | St.Dev | p25 | Median | p75 |
|---|---|---|---|---|---|---|
| $\Delta REM1$ | 654 | 0.005 | 0.093 | -0.040 | 0.002 | 0.048 |
| $\Delta REM2$ | 654 | 0.008 | 0.111 | -0.046 | 0.005 | 0.061 |
| $\Delta REM1\_R$ | 654 | 0.006 | 0.091 | -0.040 | 0.007 | 0.051 |
| $\Delta REM\_OANCF$ | 654 | 0.024 | 0.070 | -0.016 | 0.024 | 0.063 |
| $\Delta REM\_DISX$ | 654 | -0.052 | 0.199 | -0.139 | -0.039 | 0.044 |
| $\Delta REM\_PROD$ | 654 | 0.028 | 0.196 | -0.075 | 0.011 | 0.102 |

**Panel B: Basic firm characteristics**

|  | N | Mean | St.Dev | p25 | Median | p75 |
|---|---|---|---|---|---|---|
| REM1 | 654 | 0.027 | 0.226 | -0.086 | 0.016 | 0.119 |
| REM2 | 654 | 0.079 | 0.380 | -0.114 | 0.052 | 0.224 |
| REM1_R | 654 | 0.044 | 0.208 | -0.062 | 0.040 | 0.145 |
| SIZE | 654 | 7.812 | 2.053 | 6.348 | 7.954 | 9.288 |
| LEV | 654 | 0.518 | 0.210 | 0.395 | 0.504 | 0.607 |
| TOBINQ | 654 | 1.539 | 0.727 | 1.079 | 1.338 | 1.755 |
| ROA | 654 | 0.036 | 0.079 | 0.017 | 0.045 | 0.071 |
| PPENT | 654 | 0.360 | 0.179 | 0.220 | 0.339 | 0.484 |
| SGR | 654 | 0.067 | 0.170 | -0.008 | 0.054 | 0.126 |
| VOL | 654 | 0.098 | 0.058 | 0.062 | 0.087 | 0.122 |

**Panel C: Smoothness test: Vote margin within [-0.02, 0.02]**

|  | Pass = 0 | Pass = 1 | Difference around cutoff | T-statistics |
|---|---|---|---|---|
| SIZE | 7.894 | 7.562 | 0.332 | 0.513 |
| LEV | 0.472 | 0.52 | -0.048 | -0.681 |
| TOBINQ | 1.618 | 1.472 | 0.145 | 0.789 |
| ROA | 0.037 | 0.035 | 0.001 | 0.068 |
| PPENT | 0.347 | 0.383 | -0.037 | -0.687 |
| SGR | 0.045 | 0.113 | -0.068 | -1.026 |
| VOL | 0.114 | 0.112 | 0.002 | 0.104 |

*Note*: This table shows the summary statistics of the sample used in the RD analysis. Panel A reports summary statistics of REM measures. *REM_PROD* measures REM in production. *REM_DISX* measures REM in discretionary expenses. *REM_OANCF* measures REM in operating cash flow. *REM1* is as defined by [14]. *REM2* is as defined by [9]. *REM1_R* is also defined by [14] with the abnormal operating cash flow measure calculated as per [45]. Panel B shows the summary statistics of firm fundamentals. Panel C reports the results of smoothness tests on firm fundamentals shown in Panel B. The bandwidth is selected according to [48]'s mean squared error optimal bandwidth selection method.

average, decrease $\Delta REM1$ by 0.1. Because the standard deviation of $\Delta REM1$ in our sample is 0.09, the main effect is not only statistically significant but also economically meaningful. Overall, this evidence supports $H_1$, which predicts a negative effect of labor unionization at a firm on its REM.

## Robustness tests

We conduct two sets of tests to examine the robustness of the main effect reported by the local RD analysis.

First, we examine whether the main effect is driven by the selection of bandwidth. Panel A of Table 3 reports the LATEs for different lengths of optimal bandwidth based on the asymptotic mean squared error and across different kernel functions with second-order polynomials ([48]). The main effect holds in each combination.

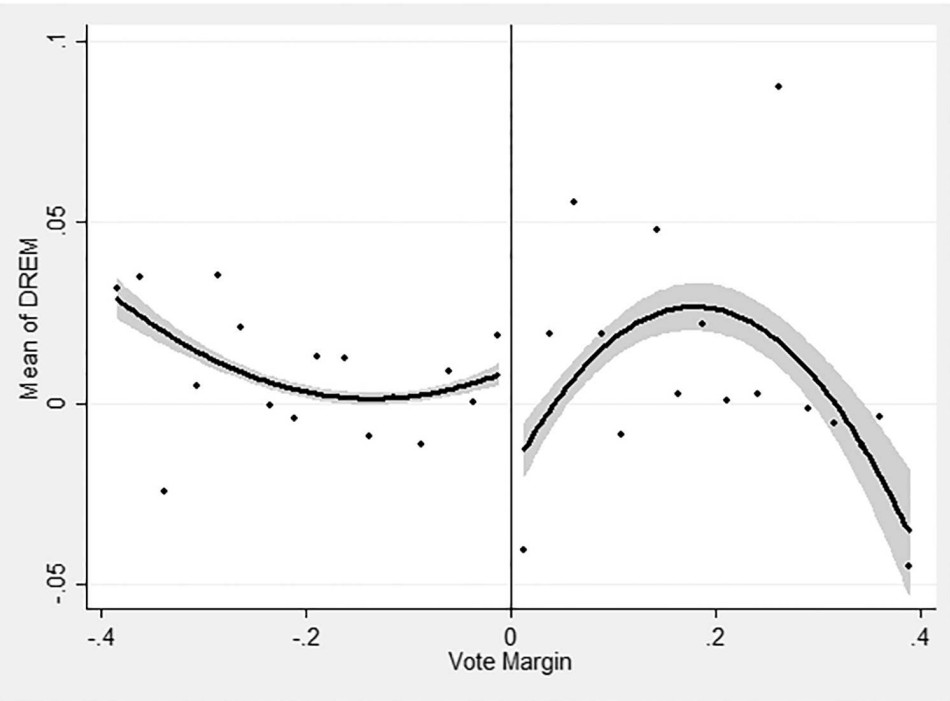

**Fig 3. RD Plot on REM.** *Note*: This figure reports regression discontinuity plots using a fitted quadratic polynomial estimate with a 95% confidence interval around the fitted value. The x-axis shows the vote margin. The y-axis shows the change in REM after labor elections. The vote margin is the difference between the vote rate and the passing threshold of 0.5. The model is estimated with polynomial order 2 and the triangular kernel.

Fig 4 reports further evidence on RD estimates with alternative bandwidths using local linear regressions with the choice of optimal bandwidth based on [48]'s method. The x-axis represents the percentage of IK optimal bandwidth. The y-axis shows the estimation results. All tests estimate the model with polynomial order 2 and the triangular kernel. As Fig 4 depicts, the RD estimates for REM are relatively stable across different lengths of optimal bandwidth. The results are similar when we use other polynomial orders and kernel functions.

Second, we test whether our results are robust to three alternate methods of bandwidth selection: (1) asymmetric mean square error (*AsyMSE*) by [48], (2) symmetric coverage error rate (*CER*), and (3) asymmetric coverage error rate (*AsymCSR*) by [53]. Panel B of Table 3 reports the LATEs when using alternate bandwidth selection methods across different orders of polynomials with the triangular kernel. As Panel B of Table 3 shows, the results for both measures of REM are similar to those of baseline regressions.

Panels C-D of Table 3 report the results for alternate REM measures. Columns (1)-(3) of Panel C show results for [45]'s REM measure (*ΔREM1_R*), while columns (4)-(6) report the results when the REM measure in the following year is used as DVs ($REM1_{t+1}$). Panel D reports the results when [3]'s REM measure and its value in the following year (*ΔREM1* and $REM1_{t+1}$) are used as DVs. The main effect is robust in these tests.

## Placebo tests

Following extant research (e.g., [11, 52]), we conduct two placebo tests to verify whether our main effect is driven by mechanical reasons. First, we replicate our baseline local RD analysis 1,000 times under artificial cutoffs between 0 and 1, excluding 0.5. All tests use polynomial

**Table 2. Local RD analysis.**

**Panel A: Kernel = Triangular**

| | DV = ΔREM1 | | | | | |
|---|---|---|---|---|---|---|
| | (1) | (2) | (3) | (4) | (5) | (6) |
| *Unionization* | -0.079*** | -0.115*** | -0.120*** | -0.075*** | -0.088*** | -0.094*** |
| | (-2.935) | (-3.028) | (-3.006) | (-2.813) | (-2.778) | (-2.829) |
| Polynomial | 1 | 2 | 3 | 1 | 2 | 3 |
| Vote Range | [-0.109, 0.109] | [-0.124, 0.124] | [-0.191, 0.191] | [-0.091, 0.091] | [-0.134, 0.134] | [-0.199, 0.199] |
| Effective Observations: Right | 158 | 185 | 292 | 127 | 200 | 304 |
| Effective Observations: Left | 103 | 109 | 140 | 88 | 110 | 144 |
| Control Variables | NO | NO | NO | Yes | Yes | Yes |
| Observations | 654 | 654 | 654 | 654 | 654 | 654 |

**Panel B: Kernel = Epanechnikov**

| | DV = ΔREM1 | | | | | |
|---|---|---|---|---|---|---|
| | (1) | (2) | (3) | (4) | (5) | (6) |
| *Unionization* | -0.077*** | -0.112*** | -0.118*** | -0.070*** | -0.083*** | -0.094*** |
| | (-2.875) | (-2.971) | (-3.014) | (-2.685) | (-2.700) | (-2.791) |
| Polynomial | 1 | 2 | 3 | 1 | 2 | 3 |
| Vote Range | [-0.101, 0.101] | [-0.118, 0.118] | [-0.188, -0.188] | [-0.090, 0.090] | [-0.133, 0.133] | [-0.189, 0.189] |
| Effective Observations: Right | 144 | 171 | 286 | 125 | 199 | 288 |
| Effective Observations: Left | 99 | 108 | 139 | 88 | 110 | 139 |
| Control Variables | NO | NO | NO | Yes | Yes | Yes |
| Observations | 654 | 654 | 654 | 654 | 654 | 654 |

**Panel C: Kernel = Uniform**

| | DV = ΔREM1 | | | | | |
|---|---|---|---|---|---|---|
| | (1) | (2) | (3) | (4) | (5) | (6) |
| *Unionization* | -0.081*** | -0.103*** | -0.107*** | -0.056** | -0.083*** | -0.096*** |
| | (-2.761) | (-2.851) | (-2.899) | (-2.417) | (-2.690) | (-2.883) |
| Polynomial | 1 | 2 | 3 | 1 | 2 | 3 |
| Vote Range | [-0.079, 0.079] | [-0.113, 0.113] | [-0.197, 0.197] | [-0.100, 0.100] | [-0.119, 0.119] | [-0.188, 0.188] |
| Effective Observations: Right | 108 | 161 | 301 | 141 | 175 | 286 |
| Effective Observations: Left | 82 | 108 | 142 | 97 | 108 | 139 |
| Control Variables | NO | NO | NO | Yes | Yes | Yes |
| Observations | 654 | 654 | 654 | 654 | 654 | 654 |

*Note*: This table presents the local average treatment effects (LATEs) of the local RD analysis. The DV is the REM measure defined by [20]. The LATE of labor unionization on REM is estimated by the following model: $\Delta REM_{i,t+1} = \alpha + \beta Unionization_{it} + F_r(X_{it}, \gamma_r) + F_l(X_{it}, \gamma_l) + \varepsilon_{i,t+1}$ where *REM* denotes the measure of REM; X denotes the voting margin, which is the difference between the voting rate and passing threshold (0.5 in this paper). Columns (1)-(3) show the results for *REM* without control variables. Columns (4)-(6) show the results for REM with control variables. The bandwidth is selected according to [48]'s mean squared error optimal bandwidth selection method. Panels A-C show the results when the Triangular kernel, Epanechnikov kernel, and Uniform kernel are used, respectively.

*, **, and *** indicate significance at the 10%, 5%, and 1% levels, respectively. z-statistic values are shown in parentheses.

order 2 and the triangular kernel. Fig 5A reports the histogram of the RD estimates and includes a dashed vertical line that represents the RD estimate at the true threshold. As shown, the pseudo estimates are all around 0, meaning that the treatment effect of unionization on REM is absent at artificially chosen vote thresholds.

Second, we estimate a placebo RD regression in which we replace the DV with the following seven firm-specific control variables (one at a time): firm size, leverage, Tobin's Q, return of assets, collateral, sales growth rate, and firm risks. Fig 5B and Table 4 reports the RD estimates

**Table 3. Robustness tests.**

**Panel A: Different lengths of optimal bandwidth (P = 2)**

| | DV = $\Delta REM1$ | | | | | |
|---|---|---|---|---|---|---|
| | **(1)** | **(2)** | **(3)** | **(4)** | **(5)** | **(6)** |
| **Kernel** | **50%BW** | **75%BW** | **80%BW** | **120%BW** | **125%BW** | **150%BW** |
| Triangular | -0.137*** | -0.118*** | -0.115*** | -0.095*** | -0.093*** | -0.090*** |
| | (-3.106) | (-3.009) | (-2.973) | (-2.840) | (-2.831) | (-2.937) |
| Epanechnikov | -0.131*** | -0.114*** | -0.110*** | -0.092*** | -0.090*** | -0.088*** |
| | (-2.936) | (-2.897) | (-2.867) | (-2.775) | (-2.765) | (-2.913) |
| Uniform | -0.120*** | -0.099** | -0.113*** | -0.096*** | -0.096*** | -0.089*** |
| | (-2.597) | (-2.505) | (-2.913) | (-2.794) | (-2.902) | (-2.879) |

**Panel B: Bandwidths from alternative selection methods (Kernel = Triangular)**

| | DV = $\Delta REM1$ | | |
|---|---|---|---|
| | **(1)** | **(2)** | **(3)** |
| **Poly. Order** | **P = 1** | **P = 2** | **P = 3** |
| AsymMSE | -0.076*** | -0.089*** | -0.100*** |
| | (-2.797) | (-2.728) | (-2.760) |
| CER | -0.085*** | -0.100*** | -0.104*** |
| | (-2.878) | (-2.917) | (-2.860) |
| AsymCER | -0.085*** | -0.103*** | -0.115*** |
| | (-2.872) | (-2.882) | (-2.984) |

**Panel C: Alternate DV: $\Delta REM1\_R$ & $REM1_{t+1}$**

| | DV = | | | | | |
|---|---|---|---|---|---|---|
| | $\Delta REM1\_R$ | | | $REM1_{t+1}$ | | |
| | **(1)** | **(2)** | **(3)** | **(4)** | **(5)** | **(6)** |
| *Unionization* | -0.113*** | -0.119*** | -0.124*** | -0.130** | -0.153** | -0.170** |
| | (-4.019) | (-3.791) | (-3.615) | (-2.245) | (-2.406) | (-2.331) |
| Polynomial | 1 | 2 | 3 | 1 | 2 | 3 |
| Kernel | Triangular | Triangular | Triangular | Triangular | Triangular | Triangular |
| Vote Range | [-0.078, 0.078] | [-0.135, 0.135] | [-0.186, 0.186] | [-0.072, 0.072] | [-0.127, 0.127] | [-0.153, 0.153] |
| Effective Observations: Right | 104 | 199 | 282 | 95 | 189 | 235 |
| Effective Observations: Left | 80 | 110 | 139 | 76 | 110 | 118 |
| Control variables | Yes | Yes | Yes | Yes | Yes | Yes |
| Observations | 654 | 654 | 654 | 654 | 654 | 654 |

**Panel D: Alternate DV: $\Delta REM2$ & $REM2_{t+1}$**

| | DV = | | | | | |
|---|---|---|---|---|---|---|
| | $\Delta REM2$ | | | $REM2_{t+1}$ | | |
| | **(1)** | **(2)** | **(3)** | **(4)** | **(5)** | **(6)** |
| *Unionization* | -0.015 | -0.001 | -0.028 | -0.156* | -0.187* | -0.205* |
| | (-0.414) | (-0.031) | (-0.558) | (-1.759) | (-1.864) | (-1.770) |
| Polynomial | 1 | 2 | 3 | 1 | 2 | 3 |
| Kernel | Triangular | Triangular | Triangular | Triangular | Triangular | Triangular |
| Vote Range | [-0.100, 0.100] | [-0.172, 0.172] | [-0.160, 0.160] | [-0.073, 0.073] | [-0.120, 0.120] | [-0.148, 0.148] |
| Effective Observations: Right | 144 | 269 | 248 | 95 | 176 | 224 |
| Effective Observations: Left | 99 | 128 | 123 | 76 | 108 | 115 |
| Control variables | Yes | Yes | Yes | Yes | Yes | Yes |

*(Continued)*

| Observations | 654 | 654 | 654 | 654 | 654 | 654 |
|---|---|---|---|---|---|---|

*Note*: This table shows the results of robustness tests of local RD analysis. Panel A reports the local average treatment effects when adopting different lengths of optimal bandwidth based on the asymptotic mean squared error optimal bandwidth selection method across different kernel functions with polynomials order 2. Panel B reports the local average treatment effects when using alternative bandwidth selection methods across different orders of polynomials with the Triangular kernel. The selection methods include asymmetric mean square error (*AsyMSE*) by [48], symmetric coverage error rate (*CER*), and asymmetric coverage error rate (*AsymCSR*) by [53]. Panel C shows the results for $\Delta REM\_R$ and $REM1_{t+1}$, respectively. Panel D shows the results for $\Delta REM2$ and $REM2_{t+1}$, respectively.

*, **, and *** indicate significance at the 10%, 5%, and 1% levels, respectively. z-statistic values are shown in parentheses.

with these alternate DVs. As shown, the pseudo estimates are all centered at 0 in the figure and there is no statistically significant relation between labor unionization and these alternate DVs. These results suggest the absence of a treatment effect of unionization on these seven firm fundamentals. In sum, these placebo tests suggest that the main effect reported by the local RD analysis comes from the discontinuity in the labor election voting around the 50% passing threshold rather than mechanical reasons.

## Heterogeneity tests

### Right-to-work (RTW) laws

Per extant research, right-to-work legislation increases unionized employees' need to bargain with employers due to the potential free-rider problem ([1, 11]). Therefore, if $H_2$ is correct and the decrease in REM after the passage of the labor election comes from increased labor power instead of unobservable reasons, firms headquartered in states that have adopted RTW are more likely to lower REM in the aftermath of labor unionization. Specifically, we define a dummy variable, *RTW*, which equals 1 for a firm that was headquartered in an RTW-state

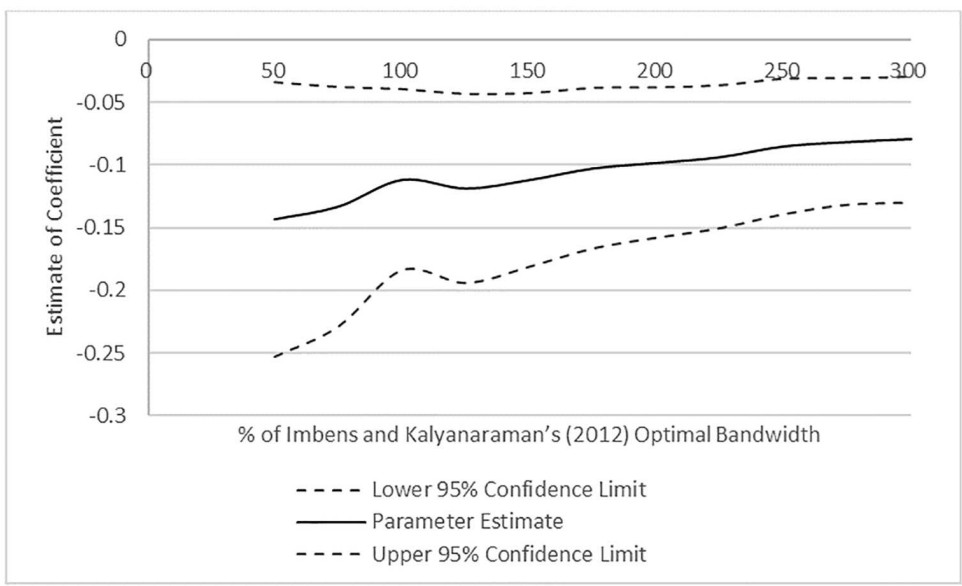

**Fig 4. RD bandwidths.** *Note*: This figure reports RD estimates with alternative bandwidths using local linear regressions with the choice of optimal bandwidth based on [46]. The x-axis represents the percentage of IK optimal bandwidth. The y-axis shows the coefficient estimates. The model is estimated with polynomial order 2 and the triangular kernel.

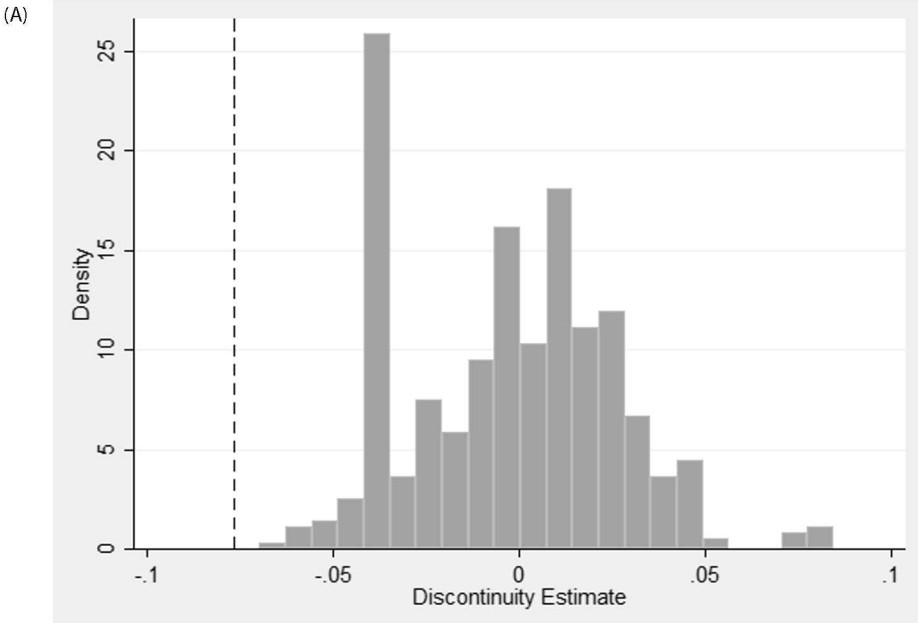

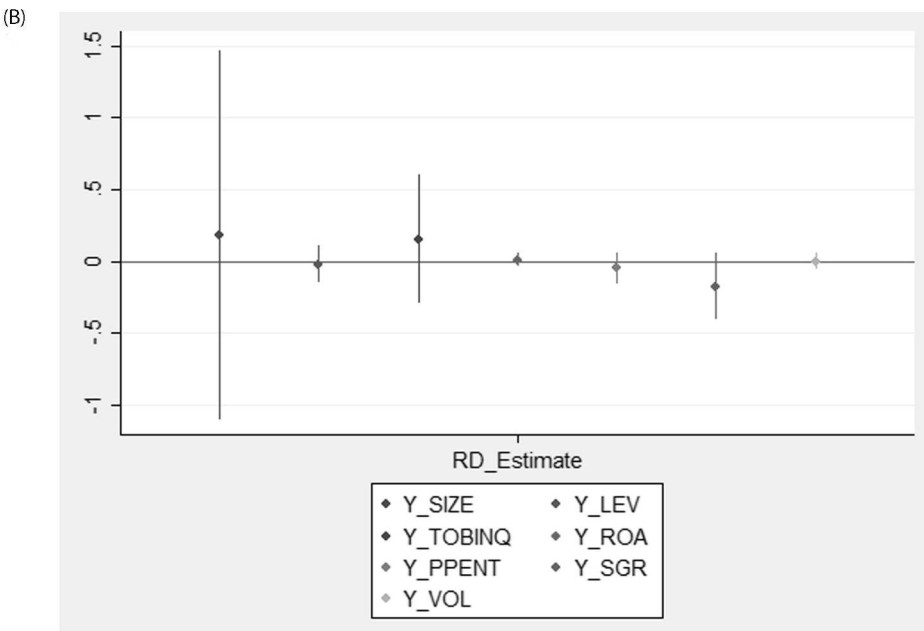

**Fig 5. Placebo tests.** Panel A: Alternate Cutoff Points. Panel B: Alternate DVs. *Note*: This figure reports RD estimates from placebo tests with alternate cutoffs and DVs. The x-axis represents the RD estimates from a placebo test under artificial cutoffs between 0 and 1 other than 0.5. The y-axis shows the fraction of the estimates. All tests estimate the model with polynomial order 2 and the triangular kernel. Panel A reports the histogram of the RD estimates with alternate cutoffs, and the vertical line shows the actual value in the baseline regressions. Panel B reports the RD estimates using control variables as DVs. The vote margin is the difference between the vote rate and the passing threshold of 0.5.

**Table 4. Placebo test: Alternate DVs.**

| | DV = | | | | | | |
|---|---|---|---|---|---|---|---|
| | *SIZE* | *LEV* | *TOBINQ* | *ROA* | *PPENT* | *SGR* | *VOL* |
| | (1) | (2) | (3) | (4) | (5) | (6) | (7) |
| *Unionization* | 0.085 | -0.022 | 0.183 | 0.011 | -0.045 | -0.131 | -0.004 |
| | (0.113) | (-0.303) | (0.706) | (0.444) | (-0.722) | (-1.528) | (-0.182) |
| Polynomial | 2 | 2 | 2 | 2 | 2 | 2 | 2 |
| Kernel | Triangular | Triangular | Triangular | Triangular | Triangular | Triangular | Triangular |
| Vote Range | [-0.165, 0.165] | [-0.163, 0.163] | [-0.154, 0.154] | [-0.154, 0.154] | [-0.156, 0.156] | [-0.125, 0.125] | [-0.211, 0.211] |
| Effective Observations: Right | 259 | 255 | 237 | 236 | 241 | 187 | 315 |
| Effective Observations: Left | 126 | 125 | 118 | 118 | 120 | 109 | 150 |
| Control Variables | Yes | Yes | Yes | Yes | Yes | Yes | Yes |
| Observations | 654 | 654 | 654 | 654 | 654 | 654 | 654 |

*Note*: This table presents the results of the placebo test where the DVs are replaced with control variables. Columns (1)-(7) report results for the DVs *SIZE*, *LEV*, *TOBINQ*, *ROA*, *PPENT*, *SGR*, and *VOL*, respectively. The Triangular kernel is used in the estimation.

*, **, and *** indicate significance at the 10%, 5%, and 1% levels, respectively. z-statistic values are shown in parentheses.

when the labor election took place, and 0 otherwise. The right-to-work states include Alabama, Arkansas, Arizona, Florida, Georgia, Idaho, Indiana, Iowa, Kansas, Louisiana, Michigan, Mississippi, Nebraska, Nevada, North Carolina, North Dakota, Oklahoma, South Carolina, South Dakota, Tennessee, Texas, Virginia, Utah, Wyoming, Wisconsin, and West Virginia. And the timeline of when state Right to Work laws were adopted was obtained from, the website of the National Right To Work Legal Defense Foundation (NRTW). Next, we estimate the RD regression for observations in the RTW-states and those in the non-RTW-states.

Table 5 reports that firms headquartered in RTW states experienced a significant decrease in REM, whereas those headquartered in non-RTW states show no significant change in REM. The difference in the coefficients of *Unionization* between RTW and non-RTW subsamples is statistically significant. This finding is consistent with $H_2$ and corroborates $H_1$.

## Benchmark beating

Next, we examine whether the managers' perceived pressure from financial analysts explains the impact of labor unionization on REM. Because a firm's failure to beat analysts' expectations decreases its reputation in the financial market (e.g., [8, 41, 43]), managers may trade off costs incurred from labor unionization for the drop in reputation. That is, managers are more likely to deflate earnings when analysts exert less pressure on a firm. To test this prediction, we separate our sample into two groups based on whether the firm beat analysts' earnings forecasts in the year the election was conducted. Specifically, we construct an index *Benchmark*, which is the analysts' earnings forecasts obtained from the Institutional Brokers' Estimate System (I/B/E/S) database to proxy for a firm's pressure from financial analysts. Firms that beat analysts' earnings forecasts (with actual earnings higher than the average analysts' forecasts) are more likely to meet investors' expectations and thus have more incentive to manipulate their earnings downwards.

As Table 6 reports, firms that beat analysts' expectations show a significant decrease in REM. The small *p*-values at the bottom of Table 6 indicate that there is a significant statistical difference in the effect of labor unions between firms with earnings higher than or lower than analysts' expectations. Because firms face greater pressure from the market when they fail to

**Table 5. Local RD: RTW legislation.**

| | Right-to-work states | | | Non-Right-to-work states | | |
|---|---|---|---|---|---|---|
| | $\Delta REM1$ | | | $\Delta REM1$ | | |
| | **(1)** | **(2)** | **(3)** | **(4)** | **(5)** | **(6)** |
| *Unionization* | -0.106** | -0.140*** | -0.138** | -0.033 | -0.046 | -0.051 |
| | (-2.401) | (-2.956) | (-2.336) | (-1.132) | (-1.192) | (-1.116) |
| Polynomial | 1 | 2 | 3 | 1 | 2 | 3 |
| Kernel | Triangular | Triangular | Triangular | Triangular | Triangular | Triangular |
| Vote Range | [-0.102, 0.102] | [-0.181, 0.181] | [-0.169, 0.169] | [-0.111, 0.111] | [-0.146, 0.146] | [-0.179, 0.179] |
| Effective Observations: Right | 50 | 99 | 95 | 102 | 140 | 181 |
| Effective Observations: Left | 35 | 47 | 45 | 68 | 73 | 88 |
| Control Variables | Yes | Yes | Yes | Yes | Yes | Yes |
| Observations | 227 | 227 | 227 | 437 | 437 | 437 |
| | $b_{(1)}$-$b_{(4)}$ | | $b_{(2)}$-$b_{(5)}$ | | $b_{(3)}$-$b_{(6)}$ | |
| Bdiff | -0.062*** | | -0.082*** | | -0.091*** | |
| P-value (Bdiff) | 0.000 | | 0.000 | | 0.000 | |

*Note*: This table presents the results of subsample tests on whether the focal firm is headquartered in states with the right-to-work legislation: Alabama, Arkansas, Arizona, Florida, Georgia, Idaho, Indiana, Iowa, Kansas, Louisiana, Michigan, Mississippi, Nebraska, Nevada, North Carolina, North Dakota, Oklahoma, South Carolina, South Dakota, Tennessee, Texas, Virginia, Utah, Wyoming, Wisconsin, and West Virginia. Cases are grouped into right-to-work groups if the firms were headquartered in right-to-work states at the time when right-to-work law was enacted in those states. The DV is the change in REM measure defined by [14]. Columns (1)-(3) report results if the employers are headquartered in states *with* right-to-work laws. Columns (4)-(6) report results if the employers are headquartered in states *without* the right-to-work law. The Triangular kernel is used in the estimation.

*, **, and *** indicate significance at the 10%, 5%, and 1% levels, respectively. z-statistic values are shown in parentheses.

**Table 6. Heterogeneity tests in local RD: Benchmark beating.**

| | DV = | | | | | |
|---|---|---|---|---|---|---|
| | Beat the benchmark | | | Fail to beat the benchmark | | |
| | $\Delta REM1$ | | | $\Delta REM1$ | | |
| | **(1)** | **(2)** | **(3)** | **(4)** | **(5)** | **(6)** |
| *Unionization* | -0.142*** | -0.164*** | -0.191** | -0.032 | -0.040 | -0.046 |
| | (-2.939) | (-2.850) | (-2.636) | (-0.970) | (-1.143) | (-1.225) |
| Polynomial | 1 | 2 | 3 | 1 | 2 | 3 |
| Kernel | Triangular | Triangular | Triangular | Triangular | Triangular | Triangular |
| Vote Range | [-0.095, 0.095] | [-0.156, 0.156] | [-0.174, 0.174] | [-0.098, 0.098] | [-0.158, 0.158] | [-0.209, 0.209] |
| Effective Observations: Right | 56 | 93 | 111 | 81 | 151 | 189 |
| Effective Observations: Left | 34 | 46 | 49 | 61 | 74 | 93 |
| Control Variables | Yes | Yes | Yes | Yes | Yes | Yes |
| Observations | 256 | 256 | 256 | 398 | 398 | 398 |
| | $b_{(1)}$-$b_{(4)}$ | | $b_{(2)}$-$b_{(5)}$ | | $b_{(3)}$-$b_{(6)}$ | |
| Bdiff | -0.098*** | | -0.114*** | | -0.139*** | |
| P-value (Bdiff) | 0.000 | | 0.000 | | 0.000 | |

*Note*: This table presents the results of subsample tests according to whether the focal firm beat analysts' expectations in the year the election was conducted. *Benchmark* is the analysts' earnings forecasts obtained from the I/B/E/S database. Cases are grouped into the *"Beat the benchmark"* group if corporate actual earnings are higher than *benchmark*, while others are grouped into the "*Fail to beat the benchmark*" group. The DV is the change in REM measure defined by [14]. Columns (1)-(3) show the results when firms beat analysts' expectations. Columns (4)-(6) show the results when firms fail to beat analysts' expectations. The Triangular kernel is used in the estimation.

*, **, and *** indicate significance at the 10%, 5%, and 1% levels, respectively. z-statistic values are shown in parentheses.

meet analyst expectations (e.g., [8, 41, 43]), this evidence provides direct support for H$_3$ and corroborates the pressure hypothesis.

## Financial resources

The literature on the financial constraint (e.g. [54, 55]) has documented that firms with higher leverage or lower levels of cash flow are more likely financially constrained, lowering managerial discretion to combat labor union demands by deflating real earnings. If this argument holds, the effect of unionization on REM should be more pronounced for firms with higher leverage or lower levels of cash flow. We test this argument. Specifically, we define (1) the leverage ratio (*LEV*) as the long-term debt and debt in current liabilities scaled by the book value of total assets and (2) the cash flow ratio (*Cashflow*) as operating cash flow scaled by the book value of total assets. We re-estimate our RD regression by splitting the sample by low versus high values of leverage and cash-flow level.

As Table 7 reports, following unionization, firms with lower leverage and higher cash flow levels decrease their earnings through REM. On the other hand, there is no significant decrease in REM for firms with fewer financial resources. Small *p*-values at the bottom of Panel A and Panel B in Table 7 indicate that there are statistical differences in the coefficients of the sub-samples. This evidence supports H$_3$ by showing that the effect of labor unionization is stronger for firms that face lower pressure resulting from adequate financial resources.

## Corporate financial risks

Next, we investigate the effect of pressure emanating from two measures of corporate financial risks. If H$_3$ is correct, firms with higher risks are less motivated to decrease their earnings because higher risks increase the cost of capital ([56]). Therefore, only firms with lower risks would decrease earnings without worrying about decreasing their cost of capital. Specifically, we investigate the market risk measured by the stock return volatility (*VOL*) calculated as the standard deviation of monthly returns during the year ([57]), and the bankruptcy risk measured by [58]'s Z score (*Altman*). We split the sample into subsamples according to the risk levels and re-estimate our RD regression for each subsample.

Table 8 reports that in the aftermath of labor unionization, firms with low risks significantly decrease their earnings through REM, whereas firms with high risks generally show little decrease in REM. The *p*-values of coefficient difference tests (0.05, on average) are generally smaller than the significance level, suggesting that there are significant statistical differences in the coefficients of *Unionization* in different groups.

Overall, evidence from a series of heterogeneity tests supports H$_2$ and H$_3$ and lends further credence to H$_1$ from the perspective of labor and managerial incentives.

## Additional tests

### Channels

Next, we investigate the channels through which labor unionization impacts REM. Based on the definition of *REM1*, we examine the change in abnormal discretionary expenses (*REM_DISX*) and abnormal operating cash flow (*REM_OANCF*) after the passage of labor elections. We also examine the change in abnormal production costs (*REM_PROD*) in untabulated results, which show no significant effect of unionization on *REM_PROD*. Taken together, the negative effects of unionization on REM are more pronounced for abnormal cash flows and abnormal discretionary expenses, suggesting that cash flows and discretionary expenses may be more easily to manipulated downwards than production costs.

**Table 7. Heterogeneity tests in local RD: Financial resources.**

**Panel A: *LEV*: high versus low**

| | DV = | | | | | |
|---|---|---|---|---|---|---|
| | **high *LEV*** | | | **Low *LEV*** | | |
| | *ΔREM1* | | | *ΔREM1* | | |
| | **(1)** | **(2)** | **(3)** | **(4)** | **(5)** | **(6)** |
| *Unionization* | -0.071* | -0.065 | -0.046 | -0.050 | -0.100** | -0.105** |
| | (-1.724) | (-1.212) | (-0.706) | (-1.502) | (-2.040) | (-2.018) |
| Polynomial | 1 | 2 | 3 | 1 | 2 | 3 |
| Kernel | Triangular | Triangular | Triangular | Triangular | Triangular | Triangular |
| Vote Range | [-0.123, 0.123] | [-0.160, 0.160] | [-0.195, 0.195] | [-0.136, 0.136] | [-0.116, 0.116] | [-0.185, 0.185] |
| Effective Observations: Right | 81 | 115 | 140 | 111 | 95 | 150 |
| Effective Observations: Left | 64 | 73 | 86 | 46 | 44 | 55 |
| Control Variables | Yes | Yes | Yes | Yes | Yes | Yes |
| Observations | 327 | 327 | 327 | 327 | 327 | 327 |
| | $b_{(1)}$-$b_{(4)}$ | | $b_{(2)}$-$b_{(5)}$ | | $b_{(3)}$-$b_{(6)}$ | |
| Bdiff | 0.019 | | 0.022 | | 0.042* | |
| P-value (Bdiff) | 0.390 | | 0.130 | | 0.060 | |

**Panel B: Cash flow: high versus low**

| | **high *Cashflow*** | | | **low *Cashflow*** | | |
|---|---|---|---|---|---|---|
| | *ΔREM1* | | | *ΔREM1* | | |
| | **(1)** | **(2)** | **(3)** | **(4)** | **(5)** | **(6)** |
| *Unionization* | -0.097*** | -0.109*** | -0.113*** | -0.035 | -0.049 | -0.045 |
| | (-2.932) | (-2.485) | (-2.492) | (-0.921) | (-0.860) | (-0.657) |
| Polynomial | 1 | 2 | 3 | 1 | 2 | 3 |
| Kernel | Triangular | Triangular | Triangular | Triangular | Triangular | Triangular |
| Vote Range | [-0.116, 0.116] | [-0.156, 0.156] | [-0.241, 0.241] | [-0.099, 0.099] | [-0.116, 0.116] | [-0.156, 0.156] |
| Effective Observations: Right | 94 | 126 | 179 | 61 | 72 | 115 |
| Effective Observations: Left | 51 | 57 | 77 | 50 | 57 | 63 |
| Control Variables | Yes | Yes | Yes | Yes | Yes | Yes |
| Observations | 328 | 328 | 328 | 326 | 326 | 326 |
| | $b_{(1)}$-$b_{(4)}$ | | $b_{(2)}$-$b_{(5)}$ | | $b_{(3)}$-$b_{(6)}$ | |
| Bdiff | -0.059* | | -0.059* | | -0.064** | |
| P-value (Bdiff) | 0.060 | | 0.070 | | 0.050 | |

*Note*: This table presents the results of subsample tests according to firms' financial resources. The DV is the change in REM measure defined by [14]. Panel A reports results if the employers have a high or low leverage, where the leverage ratio (*LEV*) is defined as the long-term debt and debt in current liabilities scaled by the book value of total assets. Panel B reports results if the employers have high or low cash flow, where the cash flow ratio (*Cashflow*) is defined as the operating cash flow scaled by the book value of total assets. Election cases are assigned to the Low (High) *Lev*/*Cashflow* group for firms with leverage/cash flow levels lower (higher) than its sample median. The Triangular kernel is used in the estimation.

*, **, and *** indicate significance at the 10%, 5%, and 1% levels, respectively. z-statistic values are shown in parentheses.

Table 9 reports that these two REM components show a significantly higher change for firms that narrowly pass the election than those that narrowly fail, meaning that labor unionization decreases a firm's abnormal discretionary expenses *and* abnormal operating cash flow.

## Labor unionization and accrual-based earnings management

We provide a more comprehensive view of how labor unionization impacts firms' earnings management decisions by showing additional tests on accrual-based earnings management

**Table 8. Heterogeneity tests in local RD: Different firm risks.**

**Panel A: Return volatility: high versus low**

| | DV = | | | | | |
|---|---|---|---|---|---|---|
| | High *VOL* | | | Low *VOL* | | |
| | $\Delta REM1$ | | | $\Delta REM1$ | | |
| | **(1)** | **(2)** | **(3)** | **(4)** | **(5)** | **(6)** |
| *Unionization* | -0.025 | -0.038 | -0.060 | -0.114*** | -0.118*** | -0.125*** |
| | (-0.659) | (-0.893) | (-1.133) | (-3.207) | (-2.990) | (-2.773) |
| Polynomial | 1 | 2 | 3 | 1 | 2 | 3 |
| Kernel | Triangular | Triangular | Triangular | Triangular | Triangular | Triangular |
| Vote Range | [-0.149, 0.149] | [-0.220, 0.220] | [-0.225, 0.225] | [-0.082, 0.082] | [-0.144, 0.144] | [-0.170, 0.170] |
| Effective Observations: Right | 105 | 161 | 164 | 62 | 115 | 138 |
| Effective Observations: Left | 57 | 76 | 76 | 39 | 56 | 65 |
| Control Variables | Yes | Yes | Yes | Yes | Yes | Yes |
| Observations | 327 | 327 | 327 | 327 | 327 | 327 |
| | $b_{(1)}-b_{(4)}$ | | $b_{(2)}-b_{(5)}$ | | $b_{(3)}-b_{(6)}$ | |
| Bdiff | 0.084*** | | 0.079** | | 0.068* | |
| P-value (Bdiff) | 0.000 | | 0.040 | | 0.060 | |

**Panel B: Altman Z-score: high versus low**

| | High *Altman* | | | Low *Altman* | | |
|---|---|---|---|---|---|---|
| | $\Delta REM1$ | | | $\Delta REM1$ | | |
| | **(1)** | **(2)** | **(3)** | **(4)** | **(5)** | **(6)** |
| *Unionization* | -0.100*** | -0.115*** | -0.114** | -0.059 | -0.057 | -0.043 |
| | (-2.770) | (-2.761) | (-2.379) | (-1.603) | (-1.442) | (-0.854) |
| Polynomial | 1 | 2 | 3 | 1 | 2 | 3 |
| Kernel | Triangular | Triangular | Triangular | Triangular | Triangular | Triangular |
| Vote Range | [-0.110, 0.110] | [-0.175, 0.175] | [-0.200, 0.200] | [-0.095, 0.095] | [-0.166, 0.166] | [-0.147, 0.147] |
| Effective Observations: Right | 90 | 144 | 161 | 60 | 122 | 104 |
| Effective Observations: Left | 55 | 67 | 72 | 43 | 62 | 55 |
| Control Variables | Yes | Yes | Yes | Yes | Yes | Yes |
| Observations | 332 | 332 | 332 | 322 | 322 | 322 |
| | $b_{(1)}-b_{(4)}$ | | $b_{(2)}-b_{(5)}$ | | $b_{(3)}-b_{(6)}$ | |
| Bdiff | -0.040 | | -0.055* | | -0.063* | |
| P-value (Bdiff) | 0.120 | | 0.080 | | 0.000 | |

*Note*: This table reports the effects of labor unionization on REM across different firm risks. The DVs are the change in the REM measure defined by [14]. Panel A presents the results of subsample tests according to firms' market risk. Market risk is measured by annual stock return volatility (*VOL*). Panel B presents the results of subsample tests by firms' bankruptcy risk. Bankruptcy risk is measured by the Altman Z-score (*Altman*). Election cases are assigned to the Low (High) group for firms with risk levels lower (higher) than its sample median. The Triangular kernel is used in the estimation.

*, **, and *** indicate significance at the 10%, 5%, and 1% levels, respectively. z-statistic values are shown in parentheses.

(AEM). Following extant research, we measure a firm's AEM as discretionary accruals based on the Kothari's model that adjusts for the firm's past performance ([15, 59]), modified Jones model ([60]), and Jones model ([61]). Appendix Table A2 in S2 Appendix shows the summary statistics of these measures.

Table 10 reports that firms with labor unionization lower AEM. Together with the results of REM, this result thus demonstrates that in the aftermath of labor unionization, managers simultaneously manipulate accrual-based and real earnings downwards.

**Table 9. Channels of REM.**

**Panel A: Abnormal discretionary expenses: REM_OANCF**

| | DV = ΔREM_OANCF | | | | | | | | |
|---|---|---|---|---|---|---|---|---|---|
| | **(1)** | **(2)** | **(3)** | **(4)** | **(5)** | **(6)** | **(7)** | **(8)** | **(9)** |
| *Unionization* | 0.038** | 0.047* | 0.050 | 0.037* | 0.045** | 0.048 | 0.035* | 0.043* | 0.046* |
| | (1.969) | (1.895) | (1.635) | (1.894) | (1.993) | (1.607) | (1.839) | (1.790) | (1.679) |
| Polynomial | 1 | 2 | 3 | 1 | 2 | 3 | 1 | 2 | 3 |
| Kernel | Triangular | | | Epanechnikov | | | Uniform | | |
| Vote Range | [-0.125, 0.125] | [-0.173, 0.173] | [-0.181, 0.181] | [-0.113, 0.113] | [-0.192, 0.192] | [-0.192, 0.192] | [-0.101, 0.101] | [-0.151, 0.151] | [-0.203, 0.203] |
| Effective Observations: Right | 185 | 274 | 281 | 161 | 293 | 293 | 144 | 229 | 309 |
| Effective Observations: Left | 109 | 128 | 135 | 108 | 141 | 141 | 100 | 116 | 146 |
| Control Variables | Yes | Yes | Yes | Yes | Yes | Yes | Yes | Yes | Yes |
| Observations | 654 | 654 | 654 | 654 | 654 | 654 | 654 | 654 | 654 |

**Panel B: Abnormal discretionary expenses: REM_DISX**

| | DV = ΔREM_DISX | | | | | | | | |
|---|---|---|---|---|---|---|---|---|---|
| | **(1)** | **(2)** | **(3)** | **(4)** | **(5)** | **(6)** | **(7)** | **(8)** | **(9)** |
| *Unionization* | 0.042** | 0.051** | 0.067** | 0.037* | 0.044** | 0.065** | 0.030* | 0.055** | 0.049* |
| | (2.037) | (2.188) | (2.387) | (1.842) | (2.049) | (2.363) | (1.716) | (2.162) | (1.905) |
| Polynomial | 1 | 2 | 3 | 1 | 2 | 3 | 1 | 2 | 3 |
| Kernel | Triangular | | | Epanechnikov | | | Uniform | | |
| Vote Range | [-0.100, 0.100] | [-0.158, 0.158] | [-0.167, 0.167] | [-0.098, 0.098] | [-0.173, 0.173] | [-0.172, 0.172] | [-0.109, 0.109] | [-0.117, 0.117] | [-0.191, 0.191] |
| Effective Observations: Right | 141 | 247 | 263 | 138 | 272 | 269 | 158 | 171 | 292 |
| Effective Observations: Left | 97 | 120 | 127 | 96 | 128 | 128 | 102 | 108 | 141 |
| Control Variables | Yes | Yes | Yes | Yes | Yes | Yes | Yes | Yes | Yes |
| Observations | 654 | 654 | 654 | 654 | 654 | 654 | 654 | 654 | 654 |

*Note*: This table reports the results of channels of REM. Panel A shows the results of abnormal operating cash flow (*REM_OANCF*). Panel B shows the results of abnormal discretionary expenses (*REM_DISX*). The bandwidth is selected according to [48]'s symptomatic mean squared error optimal bandwidth selection method. The Triangular kernel is used in the estimation.

*, **, and *** indicate significance at the 10%, 5%, and 1% levels, respectively. z-statistic values are shown in parentheses.

## External validity: Global RD analysis

Because local RD analysis investigates only the change in DVs around the threshold, we next test whether the effect of labor elections holds outside of the IK bandwidth. Therefore, we implement a parametric global RD to test the external validity of local RD. Following [49], we estimate the ATE by estimating Eq (2):

$$\Delta REM_{i,t+1} = \alpha + \beta Unionization_{it} + F_r(X_{it}, \gamma_r) + F_l(X_{it}, \gamma_l)$$
$$+ \mu Controls_{it} + Industry\ FE + Year\ FE + \varepsilon_{i,t+1} \tag{2}$$

The variables in Eq (2) are the same as those in Eq (1) and we use as covariates the same seven firm-year-specific fundamentals as we use for local RD analysis. We include year- and industry-fixed effects to control for firm-invariant but time and industry-invariant

**Table 10. Labor unionization and accrual-based earnings management.**

**Panel A: Discretionary accruals with performance ([59])**

| | DV = ΔDA_KOTHARI | | | | | | | | |
|---|---|---|---|---|---|---|---|---|---|
| | **(1)** | **(2)** | **(3)** | **(4)** | **(5)** | **(6)** | **(7)** | **(8)** | **(9)** |
| *Unionization* | -0.062** | -0.059** | -0.066** | -0.059** | -0.055* | -0.063* | -0.052** | -0.049 | -0.070** |
| | (-2.388) | (-2.010) | (-1.978) | (-2.277) | (-1.849) | (-1.838) | (-2.192) | (-1.579) | (-1.968) |
| Polynomial | 1 | 2 | 3 | 1 | 2 | 3 | 1 | 2 | 3 |
| Kernel | | Triangular | | | Epanechnikov | | | Uniform | |
| Vote Range | [-0.076, 0.076] | [-0.139, 0.139] | [-0.167, 0.167] | [-0.072, 0.072] | [-0.130, 0.130] | [-0.157, 0.157] | [-0.071, 0.071] | [-0.109, 0.109] | [-0.137, 0.137] |
| Effective Observations: Right | 100 | 204 | 261 | 93 | 193 | 240 | 93 | 156 | 200 |
| Effective Observations: Left | 78 | 110 | 127 | 76 | 110 | 120 | 76 | 103 | 110 |
| Control Variables | Yes | Yes | Yes | Yes | Yes | Yes | Yes | Yes | Yes |
| Observations | 654 | 654 | 654 | 654 | 654 | 654 | 654 | 654 | 654 |

**Panel B: Discretionary accruals based on the modified Jones model ([60])**

| | DV = ΔDA_MJONES | | | | | | | | |
|---|---|---|---|---|---|---|---|---|---|
| | **(1)** | **(2)** | **(3)** | **(4)** | **(5)** | **(6)** | **(7)** | **(8)** | **(9)** |
| *Unionization* | -0.064** | -0.061** | -0.065** | -0.061** | -0.057** | -0.064** | -0.056** | -0.051* | -0.058* |
| | (-2.575) | (-2.204) | (-2.053) | (-2.489) | (-2.002) | (-1.964) | (-2.491) | (-1.767) | (-1.658) |
| Polynomial | 1 | 2 | 3 | 1 | 2 | 3 | 1 | 2 | 3 |
| Kernel | | Triangular | | | Epanechnikov | | | Uniform | |
| Vote Range | [-0.078, 0.078] | [-0.140, 0.140] | [-0.158, 0.158] | [-0.076, 0.076] | [-0.130, 0.130] | [-0.150, 0.150] | [-0.069, 0.069] | [-0.113, 0.113] | [-0.123, 0.123] |
| Effective Observations: Right | 103 | 205 | 245 | 100 | 192 | 225 | 92 | 159 | 182 |
| Effective Observations: Left | 80 | 110 | 120 | 78 | 110 | 115 | 74 | 108 | 109 |
| Control Variables | Yes | Yes | Yes | Yes | Yes | Yes | Yes | Yes | Yes |
| Observations | 654 | 654 | 654 | 654 | 654 | 654 | 654 | 654 | 654 |

**Panel C: Discretionary accruals based on the Jones model ([61])**

| | DV = ΔDA_JONES | | | | | | | | |
|---|---|---|---|---|---|---|---|---|---|
| | **(1)** | **(2)** | **(3)** | **(4)** | **(5)** | **(6)** | **(7)** | **(8)** | **(9)** |
| *Unionization* | -0.066*** | -0.067*** | -0.068** | -0.064*** | -0.061** | -0.066** | -0.054* | -0.054** | -0.061* |
| | (-2.694) | (-2.593) | (-2.186) | (-2.637) | (-2.244) | (-2.056) | (-2.537) | (-1.867) | (-1.816) |
| Polynomial | 1 | 2 | 3 | 1 | 2 | 3 | 1 | 2 | 3 |
| Kernel | | Triangular | | | Epanechnikov | | | Uniform | |
| Vote Range | [-0.078, 0.078] | [-0.155, 0.155] | [-0.152, 0.152] | [-0.076, 0.076] | [-0.139, 0.139] | [-0.148, 0.148] | [-0.080, 0.080] | [-0.111, 0.111] | [-0.122, 0.122] |
| Effective Observations: Right | 103 | 238 | 233 | 100 | 204 | 222 | 106 | 157 | 179 |
| Effective Observations: Left | 80 | 120 | 117 | 78 | 110 | 115 | 82 | 105 | 109 |
| Control Variables | Yes | Yes | Yes | Yes | Yes | Yes | Yes | Yes | Yes |
| Observations | 654 | 654 | 654 | 654 | 654 | 654 | 654 | 654 | 654 |

*Note*: This table presents the local average treatment effects (LATEs) of the local RD analysis on accrual-based earnings management. The sample is the same as that used in the baseline local RD analysis. The bandwidth is selected according to [46]'s asymptotic mean squared error optimal bandwidth selection method. Panels A-C report the results for discretionary accruals based on discretionary accruals adjusted by firms' past performance ([59]), modified Jones model ([60]), and discretionary accruals based on the Jones model ([61]), respectively.

*, **, and *** indicate significance at the 10%, 5%, and 1% levels, respectively. z-statistic values are shown in parentheses.

**Table 11. External validity: Global RD analysis.**

**Panel A: REM**

| | DV = ΔREM1 | | | | | |
|---|---|---|---|---|---|---|
| | **(1)** | **(2)** | **(3)** | **(4)** | **(5)** | **(6)** |
| *Unionization* | -0.005* | -0.004* | -0.033** | -0.030*** | -0.061*** | -0.056*** |
| | (-1.782) | (-1.783) | (-2.528) | (-2.597) | (-2.800) | (-2.746) |
| Polynomial | 1 | 1 | 2 | 2 | 3 | 3 |
| Controls | No | Yes | No | Yes | No | Yes |
| Year & Industry FEs | Yes | Yes | Yes | Yes | Yes | Yes |
| Observations | 654 | 654 | 654 | 654 | 654 | 654 |

**Panel B: Channels of REM**

| | DV = | | | | | |
|---|---|---|---|---|---|---|
| | ΔREM_OANCF | | | ΔREM_DISX | | |
| | **(1)** | **(2)** | **(3)** | **(4)** | **(5)** | **(6)** |
| *Unionization* | 0.004 | 0.021** | 0.036** | 0.004 | 0.015* | 0.031** |
| | (1.533) | (1.995) | (1.987) | (1.212) | (1.890) | (2.033) |
| Polynomial | 1 | 2 | 3 | 1 | 2 | 3 |
| Controls | Yes | Yes | Yes | Yes | Yes | Yes |
| Year & Industry FEs | Yes | Yes | Yes | Yes | Yes | Yes |
| Observations | 654 | 654 | 654 | 654 | 654 | 654 |

*Note*: This table presents the results of the global regression discontinuity design. Following [49], we estimate the average treatment effect using Eq (2):

$$\Delta REM_{i,t+1} = \alpha + \beta Unionization_{it} + F_r(X_{it}, \gamma_r) + F_l(X_{it}, \gamma_l) + \mu Controls_{it} + Industry\ FE + Year\ FE + \varepsilon_{i,t+1} \qquad (2)$$

where *i* indicates a union election, *X* is the percentage of the vote in the labor election, and the DV is the change in REM measures. Panel A shows the results of unionization on real earnings management measured by [14]. Control variables are shown in Table 2. Year- and industry-fixed effects are included in the regression models. Standard errors are adjusted for heteroskedasticity and clustered by firm.

*, **, and *** indicate significance at the 10%, 5%, and 1% levels, respectively. Standard errors are shown in parentheses.

determinants of (change in) REM. We estimate SEs that are adjusted for heteroskedasticity and clustered by firm.

Table 11 reports that the coefficients of *Unionization* are negative and significant in models with alternate control variables and different orders of polynomials, meaning that the effect of labor unionization on REM exists in the parametric global RD design as well. On average, the passage of labor election leads to a 0.03 decrease for *REM1*. Considering the sample standard deviation of the DVs, this result is economically significant.

## External validity: Multivariate ordinary least squares (OLS)

We also conduct a multivariate OLS analysis based on a more comprehensive sample to examine whether the negative impact of labor unionization on REM is generalizable to a more representative sample. The multivariate OLS analysis uses two measures of labor unionization obtained from the Union Membership and Coverage dataset in Unionstats. Following extant research (e.g., [2, 47]), we use union membership and coverage density as industry-year-specific (and not firm-year-specific) measures of labor unionization. Union membership density (*UNION_IND*) is the total number of union members in the focal industry in the focal year divided by the total number of employees in the focal industry in the focal year. Union

coverage density (*UNION_COV_IND*) is the total number of employees covered by labor unions in the focal industry in the focal year divided by the total number of employees in the industry.

The following model estimates the impact of labor unionization:

$$REM_{i,t+1} = \alpha + \beta Union_{it} + \gamma Controls_{it} + Year\ FE + Firm\ FE + \varepsilon_{it+1} \qquad (3)$$

*REM* denotes measures of REM (*ΔREM1* and *ΔREM2*), and *Union* denotes measures of labor unionization (*UNION_IND* and *UNION_COV_IND*). Following extant research ([9, 16, 62]), we control for firm size (*SIZE*), leverage (*LEV*), Tobin's Q (*TOBINQ*), profitability (*ROA*), collateral (*PPENT*), sales growth (*SGR*), and return volatility (*VOL*). Year and firm fixed effects are included to capture time and corporate invariant factors. Industries are classified by two-digit SIC codes. Standard errors are adjusted for heteroskedasticity and clustered by firm. We source data for our OLS sample from multiple sources. The accounting data come from Compustat. The industry-level labor unionization data are from Union Membership and Coverage database in Unionstats. Observations included in this analysis should satisfy: (1) Book equity is positive; (2) All the variables used are available. Firms in financial (SIC code 6000–6999) and utility (SIC code 4900–4999) industries are excluded. All the continuous

**Table 12. Labor unionization and REM: Multivariate OLS analysis.**

| | (1) | (2) | (3) | (4) |
|---|---|---|---|---|
| | *ΔREM1* | | *ΔREM2* | |
| UNION_IND | -0.047** | | -0.062** | |
| | (-2.39) | | (-2.49) | |
| UNION_COV_IND | | -0.050*** | | -0.069*** |
| | | (-2.63) | | (-2.94) |
| SIZE | 0.030*** | 0.030*** | 0.023*** | 0.023*** |
| | (21.84) | (21.85) | (13.06) | (13.08) |
| LEV | 0.033*** | 0.033*** | -0.023*** | -0.023*** |
| | (5.45) | (5.45) | (-3.10) | (-3.09) |
| TOBINQ | -0.001 | -0.001 | -0.002** | -0.002** |
| | (-1.43) | (-1.43) | (-2.41) | (-2.41) |
| ROA | 0.108*** | 0.108*** | 0.124*** | 0.124*** |
| | (13.53) | (13.53) | (12.19) | (12.19) |
| PPENT | 0.045*** | 0.045*** | 0.032*** | 0.032*** |
| | (11.12) | (11.12) | (5.88) | (5.88) |
| SGR | 0.083*** | 0.083*** | 0.081*** | 0.081*** |
| | (31.54) | (31.54) | (26.27) | (26.27) |
| VOL | 0.058*** | 0.058*** | 0.069*** | 0.069*** |
| | (4.81) | (4.81) | (4.66) | (4.66) |
| Year, Firm FE | Included | Included | Included | Included |
| Observations | 92,147 | 92,147 | 92,147 | 92,147 |
| R-squared | 0.083 | 0.083 | 0.058 | 0.058 |

*Note*: This table reports the results of multivariate ordinary least squares testing the effect of labor unionization on REM measured by [14] and [9]. The industry-level labor unionization data come from the Union Membership and Coverage database in Unionstats. *UNION_IND* is the total number of union members over the total employees in the industry, and *UNION_COV_IND* is the ratio of the total number of employees covered by labor unions to the total number of employees in the industry. Control variables are defined in Appendix Table A1 in S1 Appendix. Industry and firm fixed effects are included to capture time and corporate invariant factors. Industries are classified by 2-digit SIC Classification. Standard errors are adjusted for heteroskedasticity and clustered by firm.

*, **, and *** indicate significance at the 10%, 5%, and 1% levels, respectively. Standard errors are shown in parentheses.

variables are winsorized at the 1% and 99% levels. Finally, the sample contains 92,147 observations during 1989–2021. Appendix Table A3 in S3 Appendix reports the summary statistics of this sample.

As Table 12 shows, the coefficients of interest are negative and significant for both *REM1* and *REM2* for each of the two measures of labor unionization. In economic magnitude, a one-standard-deviation increase in *UNION_IND* causes *ΔREM1* to decrease by 2.0% of its standard deviation, and causes *ΔREM2* to decrease by 2.2% of its standard deviation. Hence, the coefficients of interest for REM are not only statistically significant but also economically consequential. Similarly, the estimates of *UNION_COV_IND* are all negative and significant with economically meaningful magnitudes. These results further confirm $H_1$ by showing a negative relation between labor unionization and REM in a more representative sample. Overall, the results in this subsection support the external validity of the local RD analysis.

## Conclusion

Using close-call labor elections to identify the impact of labor unionization, we observe a significant decrease in REM for firms that narrowly pass the 50% passing threshold. The finding supports our main hypothesis that labor unionization pushes managers to deflate earnings through REM. This effect is robust to alternate measures of REM and different RD settings. Heterogeneity tests further support our prediction by showing that firms are less likely to decrease REM under lower labor union incentives and more likely to decrease REM under less external pressure on earnings. Taken together, these results support our prediction that managers intend to deteriorate earnings to avoid potential costs and uncertainties from labor unionization. Evidence from accrual-based earnings management provides a comprehensive view of how labor unionization affects a firm's earnings management decisions. Tests based on the global parametric RD and the multivariate OLS analysis provide further evidence of the external validity of the main finding from the local RD regression.

Overall, our research contributes to the literature on the economic consequences of labor unionization and determinants of a firm's REM decisions. Because a firm's REM distorts its optimal future cash flow and thus damages the firm's long-term value ([3, 4]), our finding sheds light on the real effect of increased labor power on a firm's long-run value. This finding is consistent with the literature on the bright versus dark side of labor unionization (e.g., [1, 11–13]). Meanwhile, because labor union activities significantly (1) increase a firm's labor cost in the United States, (2) cause offshoring of manufacturing, and (3) hinder foreign investment, the society needs an optimal balance between employees and employers. We leave an examination of this balance for future research.

In addition, we acknowledge that our Eq 3 controls for seven firm-year-specific covariates and fixed effects at the levels of year and firm. We considered including corporate governance variables (from BoardEx, for example), but such inclusion decreased our sample from 654 to 145 observations. Because our specification includes several regressors, a sample of 145 lacks statistical power to allow valid inference. We thus acknowledge this limitation and suggest future research to consider including corporate governance variables to further validate our results.

## Supporting information

**S1 Appendix. Appendix Table A1: Definitions and data sources of main variables.**
(DOCX)

**S2 Appendix. Appendix Table A2: Summary statistics of accrual-based earnings management measures.**
(DOCX)

**S3 Appendix. Appendix Table A3: Multivariate OLS analysis: Summary statistics.**
(DOCX)

## Author Contributions

**Conceptualization:** Vivek Astvansh, Beibei Wang, Tao Chen.

**Data curation:** Beibei Wang.

**Formal analysis:** Beibei Wang.

**Investigation:** Beibei Wang.

**Methodology:** Beibei Wang.

**Project administration:** Vivek Astvansh.

**Software:** Beibei Wang.

**Validation:** Beibei Wang.

**Visualization:** Beibei Wang.

**Writing – original draft:** Jimmy Chengyuan Qu.

**Writing – review & editing:** Vivek Astvansh, Beibei Wang, Tao Chen.

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
