## [Decision Letter · Decision Letter 0]

12 Apr 2023

PONE-D-23-07458Labor Unionization and Real Earnings Management: Evidence from Labor ElectionsPLOS ONE

Dear Dr. Astvansh,

Thank you for submitting your manuscript to PLOS ONE. After careful consideration, we feel that it has merit but does not fully meet PLOS ONE’s publication criteria as it currently stands. Therefore, we invite you to submit a revised version of the manuscript that addresses the points raised during the review process.

We look forward to receiving your revised manuscript.

Kind regards,

Yuan-Teng Hsu, Ph.D.

Academic Editor

PLOS ONE

Journal Requirements:

3. Please ensure that you include a title page within your main document. You should list all authors and all affiliations as per our author instructions and clearly indicate the corresponding author.

Additional Editor Comments:

The paper was reviewed by two reviewers who kindly invested their time on helping improve this work. Even though they have a positive comment on this paper, they still have several concerns that need to be addressed. I do share the referees’ views a lot and I am inclined to give you the opportunity to comprehensively/convincingly address their comments and suggestions to distinguish the paper from the existing literature.

Reviewers' comments:

Reviewer's Responses to Questions

**Comments to the Author**

1. Is the manuscript technically sound, and do the data support the conclusions?

Reviewer #1: Partly

Reviewer #2: Yes

2. Has the statistical analysis been performed appropriately and rigorously? 

Reviewer #1: Yes

Reviewer #2: Yes

3. Have the authors made all data underlying the findings in their manuscript fully available?

Reviewer #1: Yes

Reviewer #2: No

4. Is the manuscript presented in an intelligible fashion and written in standard English?

Reviewer #1: Yes

Reviewer #2: Yes

5. Review Comments to the Author

Reviewer #1: The authors investigate whether labor unionization is associated with real earnings management. The authors find evidence to suggest that firms with labor unionization are more likely to engage in real earnings management. It is commended that the paper is well written with high coherence in the exposition. The research is important. Below I offer my comments and suggestions to help further improve the paper.

Main concerns:

1. The authors need to justify their regression discontinuity design (RDD) better and do more diagnostic tests and robustness checks. For instance, is the fuzzy RDD or sharp RDD being used in the empirical test? It is good to do balance checks to see whether pre-treatment covariates are balanced and smooth around the discontinuity point. But the pre-treatment covariates should include common firm characteristics relating to firm size, growth, financial health, risks, performance and governance. For robustness check, run a placebo RDD regression in which the dependent variable is replaced with the pre-treatment covariate, or include the pre-treatment covariates as the control variable in the RDD regression; re-do the RDD regression by randomly pick a non-discontinuity point for the forcing variable and the key independent variable; do a test for discontinuity in the density of the forcing variable.

2. The magnitude of abnormal discretionary expense is linearly associated with, and proportional to, the extent of earnings being manipulated. However, the association between abnormal production costs and the manipulated amount of earnings is not linear nor proportional to one another, and the nonlinearity depends crucially on the amount of fixed overhead costs and the units of products produced in the current period. Therefore, please do not sum the abnormal production costs and abnormal discretionary expenditures for measuring real earnings management. Rather, do a common factor analysis to synthesize a composite measure based on the abnormal production costs and abnormal discretionary expenditures.

3. Regarding variable measurements, please refer to He et al. (2022) on how to measure the abnormal operating cash flows in a better way. Also, refer to He (2015) to use alternative measures of abnormal accruals for robustness tests. Regarding the measure of firm risks, please use earnings volatility (in lieu of Altman’s Z score and financial leverage), as the latter two measures reflect more of the needs for cash rather earnings.

4. Please maintain the same set of control variables across all the regressions in which the dependent variable is earnings management.

References:

He, G. (2015). The effect of CEO inside debt holdings on financial reporting quality. Review of Accounting Studies 20(1): 501-536.

He, G., Li, Z. & Shen, D. (2022). The role of earnings management in equity valuation. In Encyclopedia of Finance. Lee, C.F. & Lee, A.C. Springer. 2061-2094.

Reviewer #2: Referee report on “Labor Unionization and Real Earnings Management: Evidence from Labor Elections”

Summary

Using close-call labor elections data, the authors document that labor unionization has a positive effect on firms’ real earnings management, suggesting the pressure effect of increased labor power. They further find that this effect is attenuated for firms located in right-to-work states, and is strengthened when managers face higher pressure on earnings. Their findings are robust to alternative model specifications and alternative measures of real earnings management.

Comments

In overall, this manuscript is well written, the logic is clear and the empirical tests are rigorously executed. My specific comments on the paper are stated below.

1. A central assumption in the manuscript is that when the labor union is powerful, the company is under pressure to manipulate earnings upwards. However, this may not necessarily be the case. Companies also manage earnings downwards to combat labor union demands. For example, when companies have abundant earnings to report, a powerful labor union would request for wage rises for the workers, and the management are reluctant to incur additional labor costs. They would be motivated to manipulate earnings downloads to avert demands from labor unions. The authors could incorporate this possibility in their arguments to offer a more complete picture of how labor unionization affects firms’ accounting choices.

2. The main measures of real earnings management used in this manuscript follow Zang (2012) and Cohen and Zarowin (2010), both are widely used in prior literature. However, Chen, Hribar, and Melessa (2018) show that the two-stage approach of first estimating REM and then using it as a dependent variable causes second-stage coefficients and standard errors to be biased. The authors may refer to this paper for suggestions to overcome these biases.

3. In the sample selection part, it is better to include a sample formation procedure (for example, what is the initial sample size, and how the data requirements reduce the sample to a final size of 562 firm-year observations).

4. In the “Measures of Real Earnings Management” part, the sequence of the two equations is mixed. When introducing the measure of abnormal production costs, the equation and related discussions are about DISX. When introducing the measure of abnormal discretionary expenses, the equation and related discussions are about PROD.

5. In the “HETEROGENEITY TESTS” part, the measures used for right-to-work states, benchmark beating, growth opportunity and corporate risks are not explicitly explained in the main text. At least the authors should briefly introduce the variable names and descriptions for the measures before discuss the empirical results.

6. In the proxies used for external earnings pressure, the “growth opportunity” proxy is not intuitive to me. The authors argue that firms with few growth opportunities are more likely to be financially constrained, but this is not necessarily true. On the contrary, when considering the life cycle of firms, firms at their maturity stage have less growth potentials, but their operating cash flows just get to peak. Therefore, growth opportunity is a noisy proxy for external earnings pressure.

7. In the subsample analysis for Tables 4-7, for several columns, the coefficients on Unionization are both significant in two groups, and only the magnitudes of the coefficients diff. Is there a way to test the statistical difference in the coefficients in the subsamples? My concern is that there is actually no significant statistical difference in the coefficients of Unionization in the two groups.

8. In Table 11, when discussing the economic significance of the coefficients, it’s more informative to illustrate how a standard deviation change in X leads to a certain standard deviation change in Y.

9. Some typos found in the main text:

Page 7: The numbers follow Section are missing?

Page 8: Although wining the labor election increases labor power

6. PLOS authors have the option to publish the peer review history of their article (what does this mean?). If published, this will include your full peer review and any attached files.

Reviewer #1: No

Reviewer #2: No

---

## [Author Response · Author response to Decision Letter 0]

21 Jul 2023

Please read the "Response to Reviewers.docx" we have uploaded in EditorialManager.com.

---

## [Decision Letter · Decision Letter 1]

30 Aug 2023

PONE-D-23-07458R1Labor Unionization and Real Earnings Management: Evidence from Labor ElectionsPLOS ONE

Dear Dr. Astvansh,

Thank you for submitting your manuscript to PLOS ONE. After careful consideration, we feel that it has merit but does not fully meet PLOS ONE’s publication criteria as it currently stands. Therefore, we invite you to submit a revised version of the manuscript that addresses the points raised during the review process.

ACADEMIC EDITOR: I have completed my evaluation of your manuscript. The reviewers recommend reconsideration of your manuscript following minor revision and modification. I invite you to resubmit your manuscript after addressing their comments.

We look forward to receiving your revised manuscript.

Kind regards,

Yuan-Teng Hsu, Ph.D.

Academic Editor

PLOS ONE

Journal Requirements:

Additional Editor Comments:

I have completed my evaluation of your manuscript. The reviewers recommend reconsideration of your manuscript following minor revision and modification. I invite you to resubmit your manuscript after addressing their comments.

Reviewers' comments:

Reviewer's Responses to Questions

**Comments to the Author**

1. If the authors have adequately addressed your comments raised in a previous round of review and you feel that this manuscript is now acceptable for publication, you may indicate that here to bypass the “Comments to the Author” section, enter your conflict of interest statement in the “Confidential to Editor” section, and submit your "Accept" recommendation.

Reviewer #1: (No Response)

Reviewer #2: (No Response)

2. Is the manuscript technically sound, and do the data support the conclusions?

Reviewer #1: Yes

Reviewer #2: Yes

3. Has the statistical analysis been performed appropriately and rigorously? 

Reviewer #1: Yes

Reviewer #2: Yes

4. Have the authors made all data underlying the findings in their manuscript fully available?

Reviewer #1: Yes

Reviewer #2: Yes

5. Is the manuscript presented in an intelligible fashion and written in standard English?

Reviewer #1: No

Reviewer #2: Yes

6. Review Comments to the Author

Reviewer #1: I see quite a few merits and strengths with the revised paper. Well done! I knew there are other papers (e.g., Chang et al. 2022, Journal of Corporate Finance), which examine the same research question as does this paper under review. However, I realize there are substantial differences between the competing papers and the authors' paper. In my opinion, so long as the topic is important, it warrants several papers shedding light on it from different perspectives and based on different research designs/samples. I suggest that the authors just use a footnote to tell the differences between their paper and the competing papers. That done, the authors need to proof-read the paper carefully, and also make the citations consistent between the text and the reference list. Thank you for the opportunity to read this paper. It is a very good one for publication.

Reviewer #2: The authors have well addressed all my comments in the previous round. My only concern is, in this version, the authors have extended the sample period from 2018 to 2022, but now the results have been opposite. Therefore, I'm a bit concerned about the robustness of the results. The authors need to provide more justifications for the data in this version. Is the opposite result driven by intertemporal changes, or restructure of data in all periods?

7. PLOS authors have the option to publish the peer review history of their article (what does this mean?). If published, this will include your full peer review and any attached files.

Reviewer #1: **Yes: **Guanming He

Reviewer #2: No

---

## [Author Response · Author response to Decision Letter 1]

31 Aug 2023

I have uploaded "Response to Reviewer.docx" file.

---

## [Decision Letter · Decision Letter 2]

2 Oct 2023

Labor Unionization and Real Earnings Management: Evidence from Labor Elections

PONE-D-23-07458R2

Dear Dr. Astvansh,

We’re pleased to inform you that your manuscript has been judged scientifically suitable for publication and will be formally accepted for publication once it meets all outstanding technical requirements.

Kind regards,

Yuan-Teng Hsu, Ph.D.

Academic Editor

PLOS ONE

Additional Editor Comments (optional):

I have confirmed the current modified version and the response to the reviewer. I think it is suitable for publication.

Reviewers' comments:

Reviewer's Responses to Questions

**Comments to the Author**

1. If the authors have adequately addressed your comments raised in a previous round of review and you feel that this manuscript is now acceptable for publication, you may indicate that here to bypass the “Comments to the Author” section, enter your conflict of interest statement in the “Confidential to Editor” section, and submit your "Accept" recommendation.

Reviewer #1: All comments have been addressed

Reviewer #2: All comments have been addressed

2. Is the manuscript technically sound, and do the data support the conclusions?

Reviewer #1: Yes

Reviewer #2: Yes

3. Has the statistical analysis been performed appropriately and rigorously? 

Reviewer #1: Yes

Reviewer #2: Yes

4. Have the authors made all data underlying the findings in their manuscript fully available?

Reviewer #1: Yes

Reviewer #2: No

5. Is the manuscript presented in an intelligible fashion and written in standard English?

Reviewer #1: Yes

Reviewer #2: Yes

6. Review Comments to the Author

Reviewer #1: Well done! Overall, the paper is quite strong and of high quality. I recommend acceptance of it for publication.

Reviewer #2: (No Response)

7. PLOS authors have the option to publish the peer review history of their article (what does this mean?). If published, this will include your full peer review and any attached files.

Reviewer #1: **Yes: **Guanming He

Reviewer #2: No

---

## [Editor Report · Acceptance letter]

6 Feb 2024

PONE-D-23-07458R2 

PLOS ONE

Dear Dr. Astvansh, 

I'm pleased to inform you that your manuscript has been deemed suitable for publication in PLOS ONE. Congratulations! Your manuscript is now being handed over to our production team.

Kind regards, 

on behalf of

Dr. Yuan-Teng Hsu 

Academic Editor

PLOS ONE